# Late-Phase Second-Order Training

**Lukas Tatzel, Philipp Hennig & Frank Schneider**
University of Tübingen, Tübingen AI Center
Maria-von-Linden-Straße 6, Tübingen, Germany
`[lukas.tatzel | philipp.hennig | frank.schneider]@uni-tuebingen.de`

## Abstract

Towards the end of training, stochastic first-order methods such as SGD and ADAM go into diffusion and no longer make significant progress. In contrast, Newton-type methods are *highly* efficient "close" to the optimum, in the deterministic case. Therefore, these methods might turn out to be a particularly efficient tool for the final phase of training in the stochastic deep learning context as well. In our work, we study this idea by conducting an empirical comparison of a second-order Hessian-free optimizer and different first-order strategies with learning rate decays for late-phase training. We show that performing a few costly but precise second-order steps *can* outperform first-order alternatives in wall-clock runtime.

## 1 Introduction

Despite extensive research to improve first-order optimizers (see [13] for an overview), training neural networks with these methods remains expensive, tedious, and brittle. Second-order methods use curvature information in addition to gradients, and thus offer a *conceptually* different approach that leads to faster convergence in the deterministic case. However, second-order methods are rarely adopted in the high-dimensional and noisy deep learning setting. This may be due to the higher per-iteration costs, increased complexity or due to not having a readily usable PYTORCH [11] implementation at hand. Consequently, stochastic first-order methods, such as SGD [12] or ADAM [5] remain the most popular deep learning optimizers.

**Contributions:** To make second-order deep learning optimizers more accessible to the community, we develop a flexible and easy-to-use implementation of a Hessian-free (HF) optimizer [7, 9] in PYTORCH (Section 2). This implementation is available open source at `https://github.com/ltatzel/PyTorchHessianFree`. Next, we investigate the performance of the HF optimizer and SGD across training phases. Our results suggest that the advantages of second-order methods are most useful in the final phase of training (Section 3.1). Finally, we empirically compare the HF optimizer to various learning rate adaptation strategies commonly used for late-phase training. Our results show that second-order optimizers *can* outperform SGD in test set performance within the same wall-clock runtime budget (Section 3.2).

Our experiments demonstrate that hybrid approaches, which autonomously switch to second-order methods at the end of training, might be a viable option and a promising direction for future research.

## 2 An easy-to-use Hessian-free optimizer

First, we describe our Hessian-free (HF) optimizer which relies on components from [7, 9]. Algorithm 1 shows pseudocode, more detail is in Appendix A.1. We see a need for a dedicated, open-source implementation since no such optimizer is provided by the PYTORCH package, and other second-order methods such as K-FAC [8] or L-BFGS [6] deviate further from the notion of Newton steps.

Has it Trained Yet? Workshop at the Conference on Neural Information Processing Systems (NeurIPS 2022).

**Notation:** The objective function is the empirical risk $\mathcal{L}_{\mathcal{D}}(\boldsymbol{\theta}) = \frac{1}{M} \sum_{x \in \mathcal{D}} l(\boldsymbol{\theta}, x)$ over some training set $\mathcal{D} = \{x_1, \ldots, x_M\}$ of size $M$. The loss $l(\boldsymbol{\theta}, x)$ quantifies the network's performance on sample $x \in \mathcal{D}$ for the parameter vector $\boldsymbol{\theta}$.

**Algorithm:** In each step $k$, the local objective function is approximated using a quadratic model, and its minimum is estimated using conjugate gradients (CG) [4]. The CG-subroutine returns approximate solutions $\{\boldsymbol{p}_n\}$ for the quadratic model's minimum. For this, CG requires an estimate of the gradient and multiplications with the curvature matrix — both quantities are evaluated on the same mini-batch $\mathcal{B} \subset \mathcal{D}$. Following [7], we use the positive semi-definite generalized Gauss-Newton matrix (GGN) $\boldsymbol{G}$ as curvature proxy. The damping parameter $\lambda_k > 0$ artificially increases the curvature in each direction, which in turn leads to a smaller, i.e. more conservative step. The subsequent steps require an estimation of the loss for which we use a second

---

**Algorithm 1** Hessian-free optimizer

**Require:** Initial parameters $\boldsymbol{\theta}_0$, training data $\mathcal{D}$
    **for** $k = 0, 1, \ldots$ **do**
        Draw two disjoint mini-batches $\mathcal{B}, \overline{\mathcal{B}} \subset \mathcal{D}$

        Evaluate gradient and curvature on $\mathcal{B}$, i.e.
            $\boldsymbol{g} \leftarrow \nabla \mathcal{L}_{\mathcal{B}}(\boldsymbol{\theta}_k)$
            Define $\boldsymbol{B}(\boldsymbol{d}) = \boldsymbol{G}\boldsymbol{d} + \lambda_k \boldsymbol{d}$

        $\{\boldsymbol{p}_n\} \leftarrow$ `cg`$(\boldsymbol{B}, -\boldsymbol{g})$
        $\lambda_{k+1} \leftarrow$ `lm_heuristic`$(\lambda_k, \mathcal{L}_{\overline{\mathcal{B}}}(\boldsymbol{\theta}_k))$
        $\boldsymbol{p} \leftarrow$ `cg_backtracking`$(\{\boldsymbol{p}_n\}, \mathcal{L}_{\overline{\mathcal{B}}}(\boldsymbol{\theta}_k))$
        $\alpha \leftarrow$ `line_search`$(p, \mathcal{L}_{\overline{\mathcal{B}}}(\boldsymbol{\theta}_k))$

        Apply update $\boldsymbol{\theta}_{k+1} \leftarrow \boldsymbol{\theta}_k + \alpha \boldsymbol{p}$
    **end for**

---

mini-batch $\overline{\mathcal{B}} \subset \mathcal{D}$. First, $\mathcal{L}_{\overline{\mathcal{B}}}$ is used to adapt the damping constant using a Levenberg-Marquardt-style heuristic. While the approximations $\{\boldsymbol{p}_n\}$ lead to a reduction on the stochastic quadratic model, this is not necessarily the case for the empirical risk $\mathcal{L}_{\mathcal{D}}$. Thus, $\mathcal{L}_{\overline{\mathcal{B}}}$ is used for finding a suitable candidate $\boldsymbol{p} \in \{\boldsymbol{p}_n\}$ (`cg_backtracking`) and an appropriate step size $\alpha$ (`line_search`). Lastly, the update step $\boldsymbol{\theta}_{k+1} \leftarrow \boldsymbol{\theta}_k + \alpha \boldsymbol{p}$ is applied.

**Mini-batch overfitting:** Initially, we used $\mathcal{B} \equiv \overline{\mathcal{B}}$. However, we observed that evaluating the gradient, curvature, and loss on the same data leads to an overestimation of the progress of the parameter update, resulting in insufficient damping and overly large step sizes (see Appendix A.2 for details). Instead, we separate the mechanisms responsible for "suggesting" and "assessing" an update step into two *independent* instances: We use $\mathcal{B}$ with batch size $m := |\mathcal{B}|$ for the gradient and curvature estimates and use $\overline{\mathcal{B}}$ with fixed batch size $|\overline{\mathcal{B}}| = 1024$ for the damping heuristic, CG-backtracking, and line search. This technique significantly stabilized the optimizer without sacrificing performance.

## 3 Experiments

We consider three problems from the DEEPOBS benchmark [14], mostly the ALL-CNN-C net on CIFAR-100. Results on CIFAR-10 with the 3C3D net and on the SVHN data using the WIDERESNET 16-4 are discussed in Section 3.3. For all experimental details, see Appendix B.

### 3.1 Efficiency of SGD and HF during training

**The pros and cons of Newton steps:** In the deterministic setting, Newton's method is *highly* effective near the optimum [e.g. 10, Thm. 3.5]. Therefore, it seems natural to apply Newton-based methods such as HF at the end of the training process in the stochastic setting, too. [9, p. 23] suggests this but also argues hat "precise convergence is often not necessary [...], or even undesirable (due to issues of overfitting)". Moreover, the signal-to-noise ratio of the curvature may decrease during training [2], potentially making curvature-based methods unstable.

We conduct an empirical comparison between SGD and an HF optimizer to investigate how effective these strategies are in the different *phases* of optimization. Our experiment consists of two steps:

1. **Training:** First, we train the network using SGD, with the training hyperparameters taken from an existing benchmark [13] (see Table 1 in Appendix B). At ten equally spaced checkpoints during training, the network's parameters are stored. Figure 4 in Appendix B shows the learning curves and checkpoints for all test problems.

2. **Evaluation:** Next, we empirically compare SGD with the learning rate from step 1 to a HF optimizer, starting from the checkpoints we created during training. For a fair and practically meaningful comparison, each optimizer is assigned the same runtime budget: $10\%$ of the

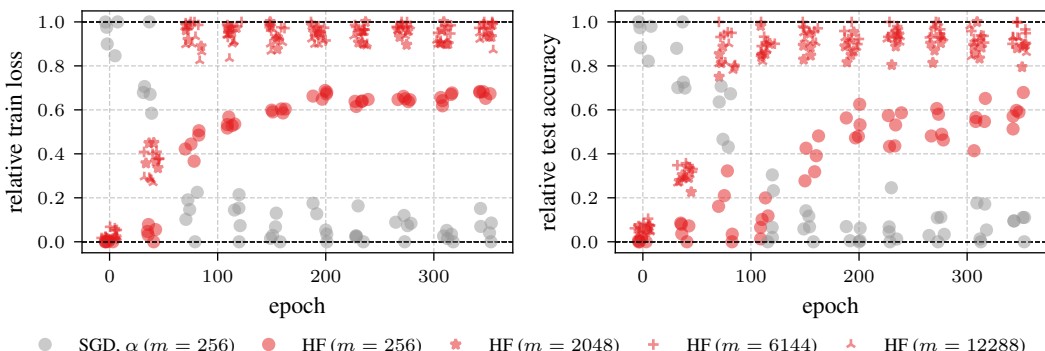

Figure 1: **Relative to SGD, HF is most performant at the end:** From each checkpoint of the ALL-CNN-C net on CIFAR-100, we resume training for a fixed runtime budget using SGD (*gray*) and HF optimizers (*red*) with different batch sizes $m$. Each setting is repeated for 5 random seeds. For each checkpoint, all runs are scored from 0 (worst) to 1 (best) according to the best train loss (*left*) and test accuracy (*right*) achieved within the budget. The markers' $x$-positions are slightly perturbed for better visibility.

time needed for training the networks in step 1. Every setting is repeated for 5 random seeds.

**Results:** For every checkpoint, we extract the best performance obtained by each optimizer within the runtime budget. These performances are then linearly transformed to range between 0 (worst) and 1 (best) to enable comparisons between checkpoints. Figure 1 shows the results for the ALL-CNN-C CIFAR-100 test problem. At the first checkpoint, SGD clearly outperforms the HF approaches. This trend reverses over the course of training for both the train loss and test accuracy. Overall, these results suggest that second-order methods are most beneficial towards the end of training.

### 3.2 Hessian-free as a late-phase optimizer

Motivated by the previous section, we focus on the use of second-order optimizers during the final phase of training. In this section, we empirically compare HF approaches to popular late-phase training strategies. The goal is to investigate, whether a few precise — but costly — second-order steps are effective when the model is close to being fully trained. To investigate this, we resume training from the very last checkpoint for $10\%$ of the total original training time and compare HF approaches using four different batch sizes to the following learning rate adjustments:

$\alpha$ For this baseline, we keep the constant learning rate that was originally used for training.

$\alpha/10$ Here, the learning rate is reduced by a factor of 10 at the checkpoint.

$\alpha_k^{\cos}$ This cosine schedule gradually decays the original learning rate to 0 over the given budget.

$\alpha_k$ Sets the learning rate dynamically according to PYTORCH's `ReduceLROnPlateau` method. This scheduler is initialized with the original learning rate and decreases it by a factor of 10 whenever the train loss cannot be improved significantly within a certain number of epochs. This parameter is chosen such that the learning rate can be reduced up to three times within the given budget (see Figure 7 for the resulting learning rate schedules).

**Results:** Figure 2 shows the result of the different late-phase training strategies for the ALL-CNN-C net on CIFAR-100 (see Figure 6 for additional test problems). We make the following observations:

- **Changing strategy pays off:** All strategies outperform the SGD baseline with a constant learning rate (shown as a solid gray line), both in terms of train loss and test accuracy.

- **Second-order methods can generalize surprisingly well:** Although SGD with the cosine schedule performs the best in terms of train loss, the HF optimizers with $m \geq 2048$ significantly outperform it in terms of test accuracy for a given wall-clock runtime budget.

- **The HF optimizer requires larger batch sizes for top peak performance:** For the HF approaches, larger batch sizes are required to reach top peak performance. But surprisingly,

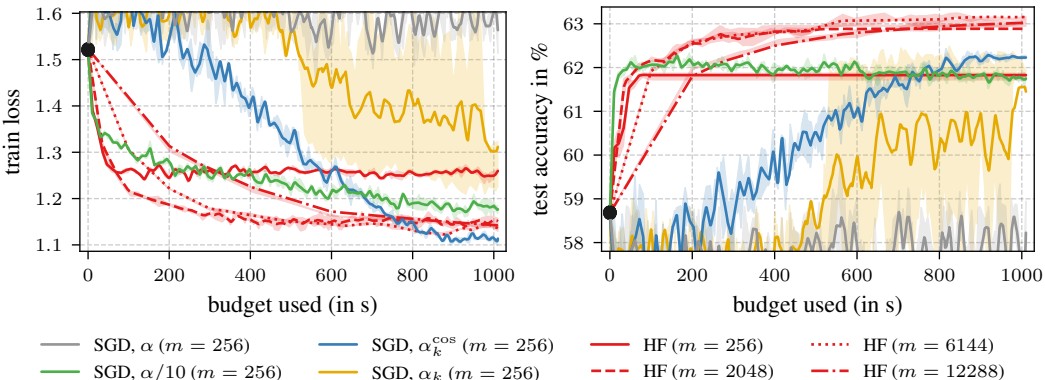

Figure 2: **In the final phase, second-order methods can outperform SGD in wall-clock time:** Comparison of different strategies for the ALL-CNN-C network on CIFAR-100 in terms of the train loss (left) and test accuracy (middle) when starting from the very last checkpoint at training epoch 349. Each setting is repeated for five different seeds. The line shows the mean over these runs, the shaded area ranges from the lower to the upper quartile.

even with the smallest batch size $m = 256$, the HF optimizer outperforms the SGD baseline (solid gray line). It is also notable that the HF approaches generally optimize quite quickly, requiring only a few steps to provide a significant performance boost, similar to the drastic reduction of the learning rate of SGD with $\alpha/10$ (green gray line).

### 3.3 Empirical results on other test problems

Most of these observations also apply to the 3C3D model on CIFAR-10, but *not* to WIDERESNET 16-4 network on SVHN. The HF optimizers again exhibit remarkable generalization ability on the 3C3D model on CIFAR-10 (top subplot in Figure 6), with HF ($m = 6144$) reaching a similar test accuracy as SGD with $\alpha/10$ despite having a significantly worse train loss. In this scenario, however, SGD with a cosine decay provides the best test accuracy. Surprisingly, however, none of the observations hold for the WIDERESNET 16-4 model on SVHN (bottom subplot in Figure 6). Here, SGD with a constant learning rate decreases the train loss significantly but the test accuracy is unaffected. We also do not observe a good performance for the HF approaches. This might indicate that the WIDERESNET 16-4 is not in the final phase of training or that the optimization problem is structurally different from the other problems. Future work could investigate automatic methods for detecting whether a model might be amenable to second-order methods, for example, by detecting if the model has entered its final phase of training. Another explanation why HF is worse on CIFAR-10 and SVHN could be that the rank of the GGN is bounded by $m \cdot C$ [2], where $C$ is the number of classes. That means, for CIFAR-100 ($C = 100$) the GGN might be more "expressive" than for the other two test problems ($C = 10$).

## 4 Conclusion

Our empirical comparison for late-phase training shows that second-order methods *can* significantly outperform first-order approaches in wall-clock runtime. Although the strong results of the HF approach with the ALL-CNN-C net on CIFAR-100 could not be observed to the same extent on the other problems (see Section 3.3), our results still show the potential of Hessian-free methods.

Our runtime comparisons depend on the hardware used and on the efficiency of the underlying implementations. The bias of the results due to the implementation turns out to be a disadvantage for the HF approach: While we use the efficient PYTORCH code for the SGD variants, our implementation of the HF optimizer is certainly less optimal. Another advantage of SGD is that today's commonly used model architectures are designed to be trained by first-order methods.

We argue that the HF approach shows potential for future research, which we want to facilitate with our open source implementation. Future work could study the individual components of the HF

optimizer, potentially leading to a simplified and more efficient method. Such work could lead to hybrid training approaches that use second-order methods for late-phase training.

## Acknowledgments and Disclosure of Funding

The authors gratefully acknowledge financial support by the European Research Council through ERC StG Action 757275 / PANAMA; the DFG Cluster of Excellence "Machine Learning - New Perspectives for Science", EXC 2064/1, project number 390727645; the German Federal Ministry of Education and Research (BMBF) through the Tübingen AI Center (FKZ: 01IS18039A); and funds from the Ministry of Science, Research and Arts of the State of Baden-Württemberg. LT is also grateful to the International Max Planck Research School for Intelligent Systems (IMPRS-IS) for support. FS is supported by funds from the Cyber Valley Research Fund. The authors are also grateful to Andres Fernandez Rodriguez, Felix Dangel and Runa Eschenhagen for feedback.

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

# Appendix

## A  Details of the Hessian-free optimizer

### A.1  Implementation details

In this section, we provide more details concerning the implementation of the Hessian-free (HF) optimizer in PYTORCH, available at `https://github.com/ltatzel/PyTorchHessianFree`. The optimizer relies on components described in [7] and [9].

**Hessian & GGN:** Our implementation allows the use of either the Hessian matrix or the generalized Gauss-Newton matrix (GGN) as a curvature matrix via the argument `curvature_opt` to the optimizer's constructor. As recommended in [7, Section 4.2] and [9, e.g. p. 10 f.], the default is the symmetric positive semidefinite GGN. To compute matrix-vector products with these matrices, we use functionality provided by the BACKPACK package [1].

**Damping:** As described in [7, Section 4.1], Tikhonov damping can be used to avoid overly large steps. Our implementation also features the Levenberg-Marquardt style heuristic for adjusting the damping parameter, which can be turned on and off via the `adapt_damping` switch. This heuristic measures the actual reduction of the objective function against the expected reduction according to the quadratic model. If this reduction ratio is small, i.e. the quadratic model does not seem trustworthy, the damping constant is increased. If it is close to one, a smaller damping is applied in the next step. By default, we use an initial damping of $1.0$.

**CG:** Our implementation of the conjugate gradient method is based on the pseudocode given in [9, Algorithm 2]. It supports preconditioning and features the termination criterion presented in [7, Section 4.4] via the argument `martens_conv_crit`. In addition, CG is terminated after at most 250 iterations as in [7] (see [9, Section 8.7]).

In principle, our implementation of the CG method can handle non-positive directional curvature, i.e. $p_i^\top A p_i \leq 0$ (note that this is a violation of the assumption that $A$ is positive definite) via the `nonpos_curv_option`-argument to the `postprocess_pAp` function. For example, it allows using the absolute value of the directional curvature — this idea is discussed in detail in [3].

As in [7, Section 4.5], we use the approximate CG solution from the last step as a starting point for the next CG run. Using the argument `cg_decay_x0` of the optimizer's constructor, this initial search direction can be scaled by a constant. The default value is $0.95$, as in [9, Section 10].

The `get_preconditioner`-method implements the preconditioner suggested in [7, Section 4.7], the diagonal of the empirical Fisher matrix. In our experiments, we do not use preconditioning.

**CG-backtracking:** When CG-backtracking is used, the CG method returns not only the final "solution" to the linear system but also intermediate approximations for a subset of the iterations. This grid of iterations is generated using the approach from [7, Section 4.6]. In a subsequent step, the set of potential update steps is searched for a suitable candidate (without having to evaluate the loss for all candidates), see [9, p. 36]. This requires evaluating the loss function possibly multiple times. CG-backtracking is used by default but can be disabled using the `use_cg_backtracking` argument.

**Line search:** Next, the proposed update step is iteratively scaled back by the line search until the loss is decreased "significantly" as determined by the Armijo condition with $c = 0.01$, see [9, Section 8.8]. By default, an initial learning rate of $1.0$ is used, which is scaled by a factor of $\beta = 0.8$ repeatedly for at most 20 times.

Our implementation performs a line search by default (which is not the case in [7]) — but it can be turned off using the `use_linesearch` switch. Empirically, it increases the stability of the HF approach, e.g. in cases where the damping determined via the LM-heuristic is insufficient.

**Handling large mini-batch sizes:** For very large mini-batch sizes, the memory required for computing the loss, gradient, and matrix-vector products might exceed the GPU's capacity. In this case, our implementation offers the method `acc_step`. It takes a list of smaller chunks of data that, when added together, gives the mini-batch that is actually to be used. It evaluates e.g. the gradient only on one list entry (i.e. one mini-batch) at a time and `accumulates` the individual gradients automatically. This iterative approach slows down the computations, but it enables us to work with very large mini-batches.

**Tests:** Our implementation also features methods that test (i) if loss and matrix-vector products behave deterministically (this is important because "inconsistent" matrix-vector products in CG would imply different quadratic models which might lead to unpredictable behavior) and (ii) if the accumulation of mini-batches over multiple chunks of data works properly (this depends on the `reduction` used by the loss function).

### A.2 Mini-batch overfitting

In this section, we provide an example of overfitting to the mini-batch and show how our approach (described in Section 2) addresses this issue. We compare the following two strategies:

1. The original version of the optimizer uses $\mathcal{B} \equiv \overline{\mathcal{B}}$ with $|\mathcal{B}| = |\overline{\mathcal{B}}| = 6144$

2. Our alternative HF approach uses two disjoint mini-batches $\mathcal{B}, \overline{\mathcal{B}}$: The first batch $\mathcal{B}$ with batch size $|\mathcal{B}| = 6144$ is used for estimating the gradient and curvature; The second batch $\overline{\mathcal{B}}$ with batch size $|\overline{\mathcal{B}}| = 1024$ is used for the damping heuristic, CG-backtracking, and the line search.

We use the ALL-CNN-C CIFAR-100 test problem and apply both strategies described above to the last checkpoint at epoch $349$. The runtime budget is set to $3600$ s (in order to provide enough time for instabilities to develop even at this large batch size) and each setting is repeated for $5$ random seeds.

Figure 3 shows the initial and final loss (before and after applying the parameter update) evaluated on $\overline{\mathcal{B}}$ for 5 runs with different random seeds. The original approach always manages to make progress

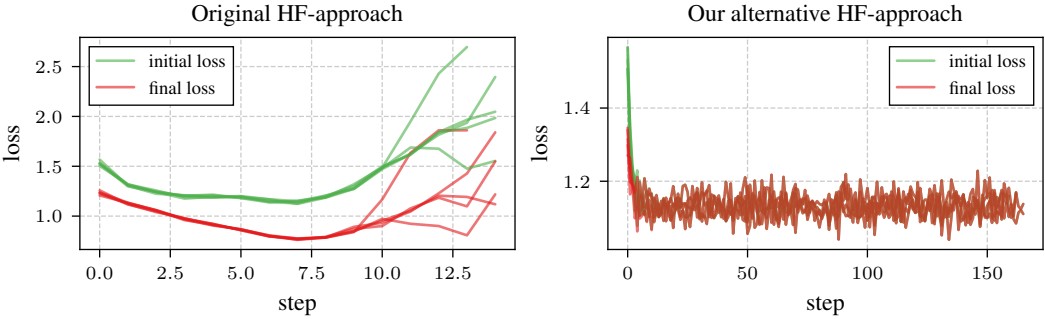

Figure 3: **Mini-batch overfitting:** Comparison of the original HF approach using $\mathcal{B} \equiv \overline{\mathcal{B}}$ with $|\mathcal{B}| = |\overline{\mathcal{B}}| = 6144$ (left) and our alternative using disjoint mini-batches with $|\mathcal{B}| = 6144$ and $|\overline{\mathcal{B}}| = 1024$ (right). Both strategies are applied to the last checkpoint at epoch $349$ of the ALL-CNN-C CIFAR-100 test problem with $3600$ s runtime budget. The initial (before applying the parameter update) and final (after applying the parameter update) loss evaluated on $\overline{\mathcal{B}}$ are shown for $5$ runs with different random seeds.

on the given mini-batch. However, a small final loss is not necessarily reflected in the initial loss of the subsequent step. This indicates overfitting to the mini-batch. The final few optimization steps even lead to an increase of the initial loss values.

Our alternative HF approach does not show such behavior: It quickly decreases the loss and keeps it stable at a relatively small loss value. Additionally, our approach performs far more steps (a factor of more than 10) due to the runtime we save by evaluating the loss on a smaller mini-batch.

## B  Experimental details

Our experiments are conducted on three classification problems from DEEPOBS [14]: 3C3D with CIFAR-10, ALL-CNN-C with CIFAR-100, and WIDERESNET 16-4 with the SVHN dataset. All test problems use the cross-entropy loss function. The experiments were executed on an NVIDIA GeForce RTX 2080 Ti GPU (11 GB).

### B.1  Training hyperparameters

The hyperparameters used for training these networks with SGD are extracted from an existing benchmark [13]. They are summarized in Table 1. Figure 4 shows the training metrics for all test problems.

Table 1: **The training hyperparameters of SGD:** We report SGD's learning rate $\alpha$, the batch size used for training, and the number of training epochs for each test problem. The learning rates are tuned for maximum accuracy on a validation dataset.

| Network | Dataset | Learning rate | Batch size | Training epochs |
|---|---|---|---|---|
| 3C3D | CIFAR-10 | $\alpha \approx 0.067966$ | $N = 128$ | 100 |
| ALL-CNN-C | CIFAR-100 | $\alpha \approx 0.171234$ | $N = 256$ | 350 |
| WIDERESNET 16-4 | SVHN | $\alpha \approx 0.025378$ | $N = 128$ | 160 |

### B.2  Performance evaluation

We empirically compare the performance of different SGD-based and HF approaches starting from each checkpoint created during training (see Appendix B.1). Each optimizer is assigned the same runtime budget: $10\%$ of the time needed for training the networks. Every setting is repeated for 5 random seeds.

#### B.2.1  Performance evaluation for training phases

Here, we investigate how the performance of the optimizers changes during different training phases. For every checkpoint, we extract the best train loss and test accuracy achieved by one of the tested optimizers within the runtime budget. We then linearly transform all observed values — separately for each checkpoint — such that the best performance (smallest loss/highest accuracy) is always at relative performance 1 and the worst performance (highest loss/smallest accuracy) is assigned a value of 0.

The results for all optimizers and test problems are shown in Figure 5. For the 3C3D CIFAR-10 and the ALL-CNN-C CIFAR-100 test problems, we observe that the relative performance of SGD with the original learning rate used for training (*gray*) decreases over the course of the training time. The alternative SGD approaches and HF optimizers show increasing relative performances over the training phases. For ALL-CNN-C CIFAR-100, the best relative performance (in terms of the test accuracy) for the second half of training is exhibited by the HF approaches with batch size $m \geq 1024$.

For the WIDERESNET 16-4 SVHN test problem, the distribution of the relative performances seems quite static over the checkpoints: SGD with the original learning rate (*gray*) remains at a relative train loss performance close to 1 over the entire training. Regarding test accuracy, SGD with the reduced learning rate (*green*) and the cosine decay (*blue*) quickly achieve high relative performances.

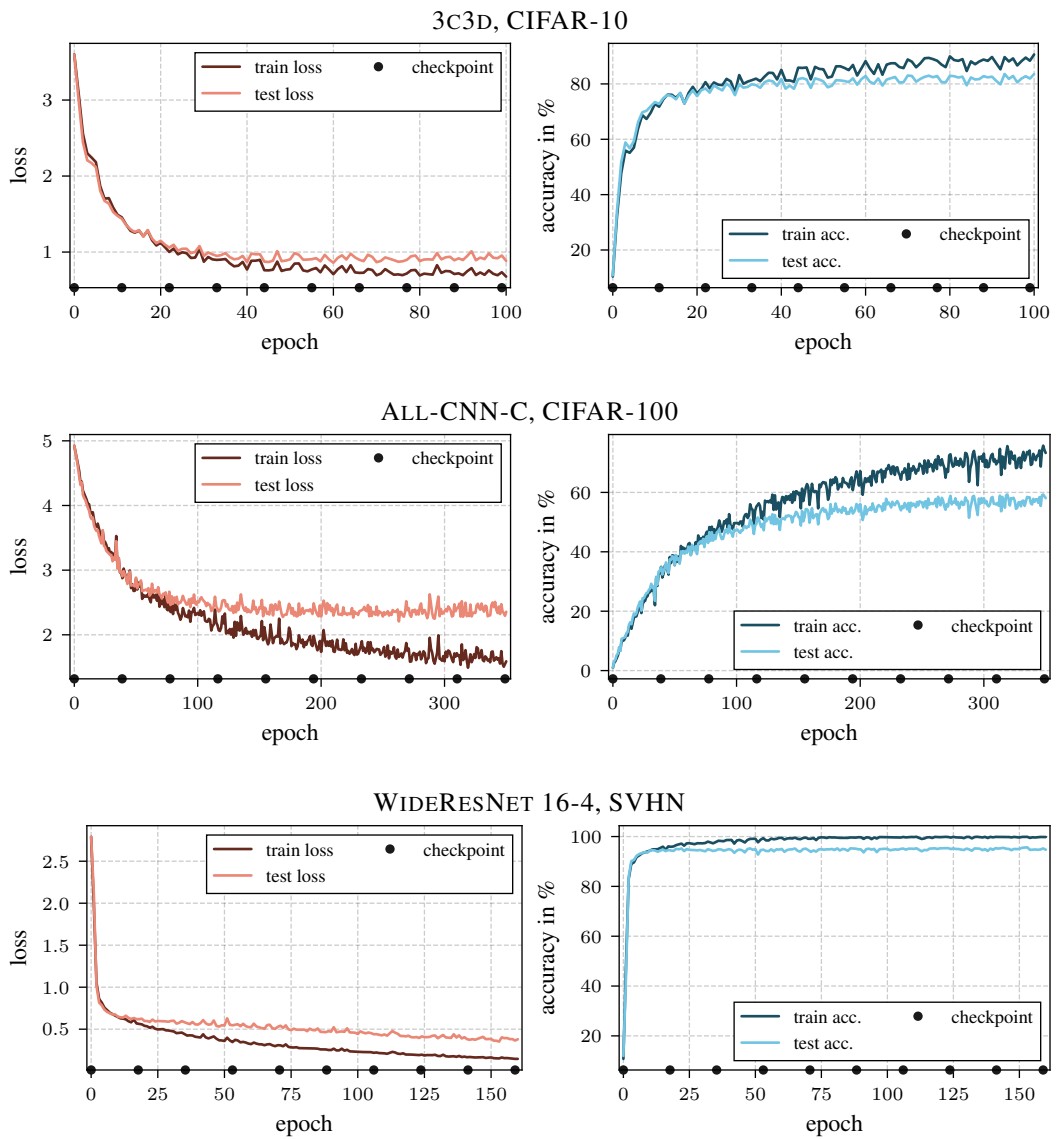

Figure 4: **Training metrics:** The loss (*left*) and accuracy (*right*) on the train and the test data are shown for all test problems. During training, we create 10 checkpoints (●).

The worst performance in terms of training loss and test accuracy is consistently exhibited by the HF approaches. It might be that we would have to run the training even longer (more than 160 epochs, see Table 1) to observe a qualitative change in the ranking of the optimizers. The model architecture (a residual network) might pose a *structurally* different optimization problem which might be one reason for the overall different observations compared to the other test problems. Our results show that HF *can* outperform SGD-based approaches in terms of the test accuracy, even with the same wall-clock runtime budget.

### B.2.2 Detailed performance evaluation for the last checkpoint

In this section, we report the optimizers' performance for the last checkpoint in more detail. Figure 6 shows the train loss (*left*) and test accuracy (*right*) over the runtime budget for the SGD-based and HF approaches starting from the very last checkpoint. Figure 7 shows the corresponding learning rate progressions for the SGD-based optimizers over the training budget.

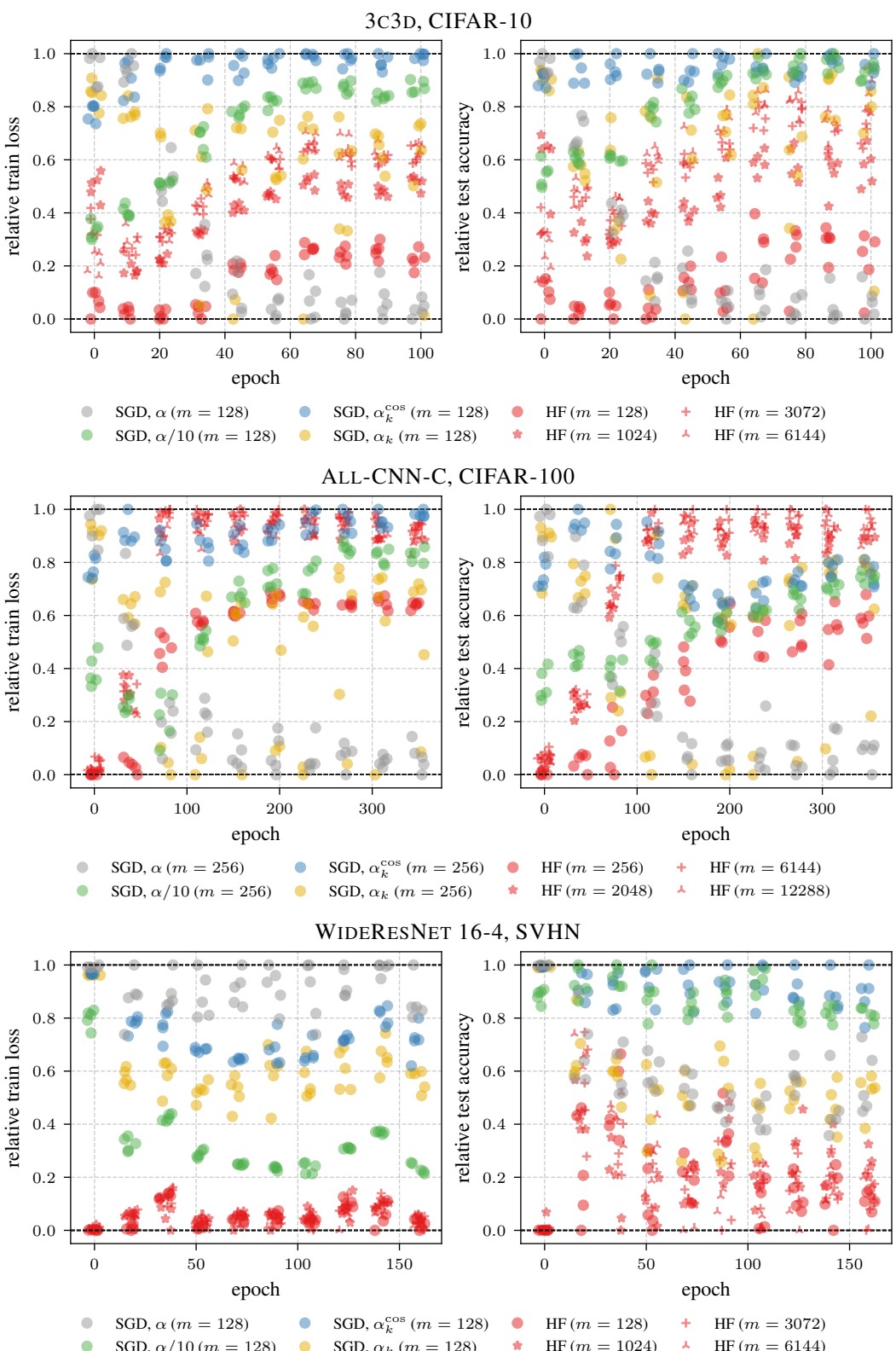

Figure 5: **Relative performance over checkpoints:** From each checkpoint, we resume training for a fixed runtime budget using SGD-based strategies or HF optimizers (*red*) with different batch sizes $m$. We report the best train loss (*left*) and test accuracy (*right*) observed within this budget. Each optimizer run is repeated for 5 random seeds. The performances are transformed linearly to range from 0 (worst) to 1 (best). The markers' $x$-positions are slightly perturbed for better visibility.

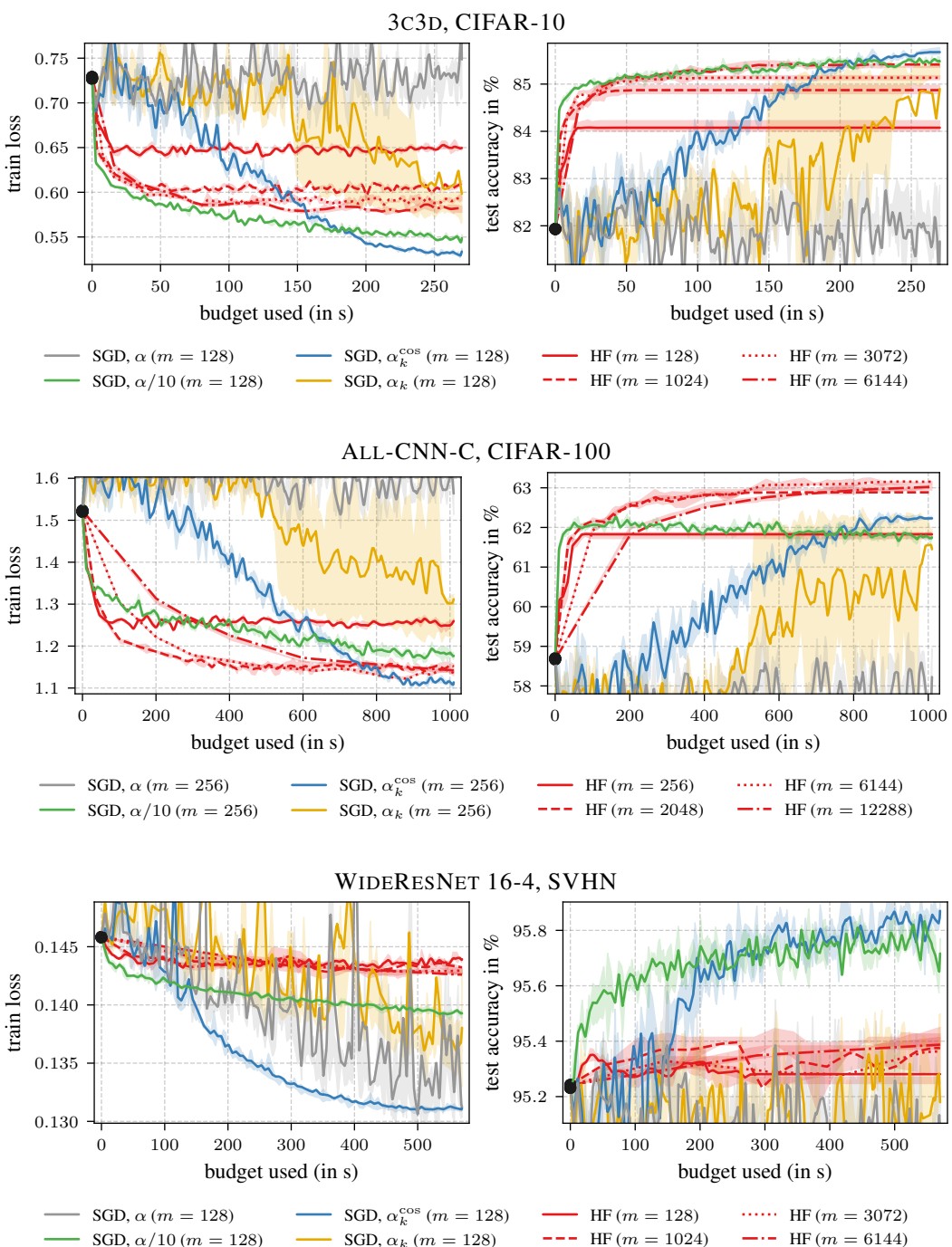

Figure 6: **Performance over budget:** Comparison of different strategies for all test problems when starting from the very last checkpoint. Each setting is repeated for five different seeds. The line shows the mean over these runs and the shaded area ranges from the lower to the upper quartile.

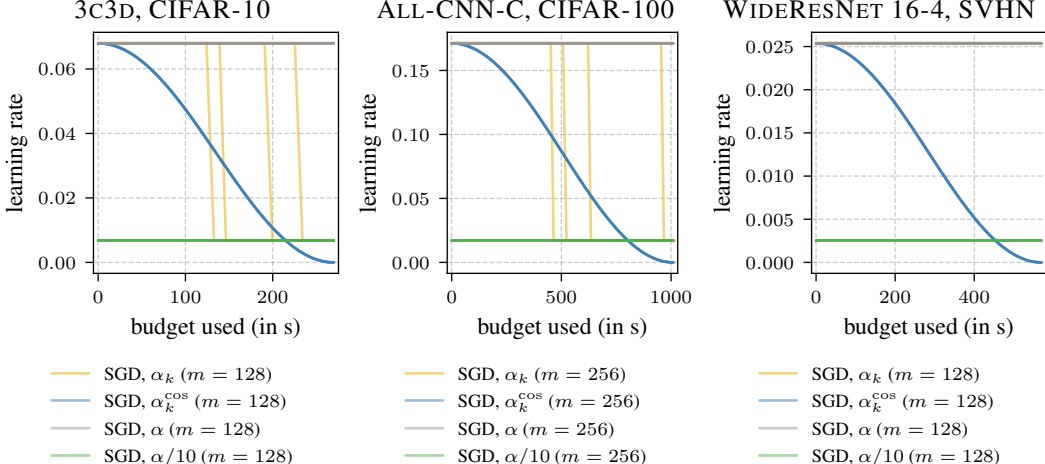

Figure 7: **Learning rates over budget:** Learning rate progressions for all test problems when starting from the very last checkpoint. Each setting is repeated for five different seeds.

