# OpenReview forum: "Late-Phase Second-Order Training"
_NeurIPS.cc/2022/Workshop/HITY — HITY Workshop NeurIPS 2022_

### Official Review · Reviewer_qt6Q · 2022-10-06
**Second-order Hessian-free methods in late-phase training**

**Rating:** 1
**Confidence:** 3

**Review:**

The authors empirically compare first-order methods with second-order Hessian-free methods in different stages of training in deep learning. The results look very promising and the content is clearly presented.

---

### Official Review · Reviewer_ShYi · 2022-10-17
**Using hessian free optimization for late stage training**

**Rating:** 1
**Confidence:** 4

**Review:**

This paper evaluates the use of a hessian free second order optimizer for late stage training where first order methods are not making much progress.  The idea makes sense and seems promising on CIFAR-100 experiments.  Results don't seem great outside of CIFAR-100 but the idea still seems worth further investigation.

---

### Decision · Program_Chairs · 2022-10-20

Accept